# Fractal and Fractional Derivative Modelling of Material Phase Change

**Harry Esmonde** 

School of Mechanical Engineering, Dublin City University, 9 Dublin, Ireland; harry.esmonde@dcu.ie

**Abstract:** An iterative approach is taken to develop a fractal topology that can describe the material structure of phase changing materials. Transfer functions and frequency response functions based on fractional calculus are used to describe this topology and then applied to model phase transformations in liquid/solid transitions in physical processes. Three types of transformation are tested experimentally, whipping of cream (rheopexy), solidification of gelatine and melting of ethyl vinyl acetate (EVA). A liquid-type model is used throughout the cream whipping process while liquid and solid models are required for gelatine and EVA to capture the yield characteristic of these materials.

**Keywords:** complex modulus; phase; frequency response function; melting; gel point; squeeze film

## 1. Introduction

Rheology is the study of the flow and deformation of liquids and soft solids and has many applications in the materials and processing industries. A large group of materials can be described as viscoelastic, that is, they exhibit behaviour that can be described by a combination of elastic and viscous stress when undergoing strain. To capture the stress/strain relationships for these materials, discrete models consisting of a combination of elastic springs (E) and viscous dampers (η) connected in different ways can be used. Typical examples would be Maxwell and Kelvin Voigt systems, shown in Figure 1.

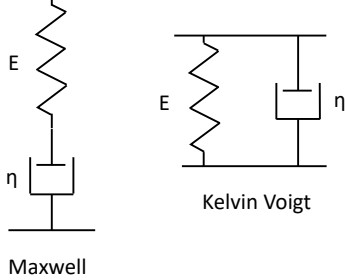

**Figure 1.** Rheological models.

The mechanical impedance $Z(s)$ in the Laplace domain relating stress to strain in each case is given by:

Kelvin Voigt:

$$Z_{KV}(s) = E + \eta s \tag{1}$$

Maxwell:

$$Z_M(s) = \frac{E\eta s}{E + \eta s} \tag{2}$$

To investigate material behaviour, dynamic material analysis is often employed where a time-varying stress or strain is applied to the material and the corresponding strain or stress is recorded from which the impedance is derived. Decomposing this into the relationship at individual frequencies, it is then possible to represent the material behaviour as a frequency response function (FRF). The FRF for both a typical Maxwell and a typical Kelvin Voigt system is shown in Figure 2 in terms of magnitude and phase, and is otherwise known as a Bode plot. It is worth mentioning here that the phase in a bode plot is the frequency-equivalent of the time shift between the output signal when compared to the input signal as the signal passes through a system. This is not to be confused with the term phase used to describe the physical form of a material, for example, liquid or solid.

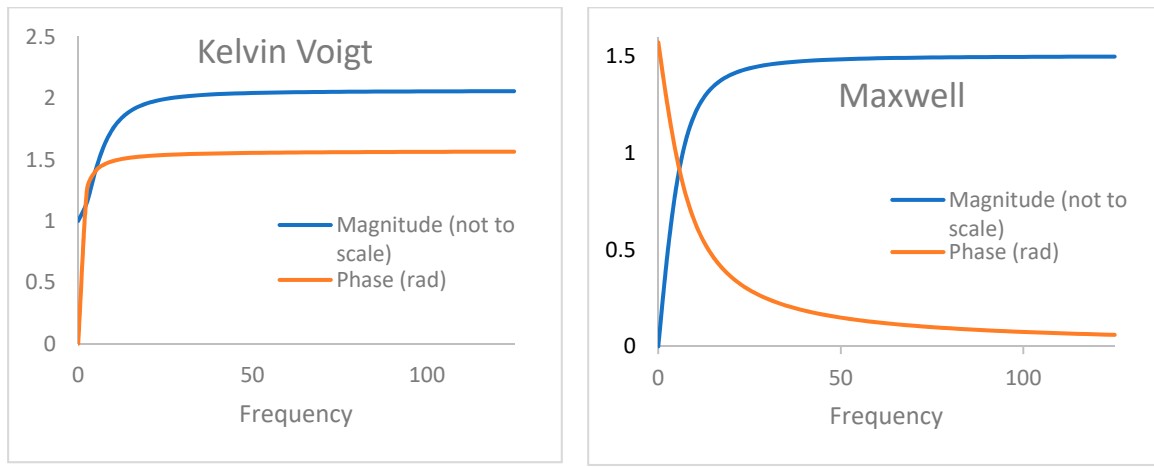

**Figure 2.** Bode plots of Maxwell and Kelvin Voigt systems.

These FRFs can be understood by replacing the Laplace operator $s$ in Equations (1) and (2) with the Fourier operator $j\omega$. Of note is the fact that the phase at low and high frequencies will tend to either zero or $\pi/2$ radians. At intermediate frequencies, the phase gradually shifts from the low-frequency level to the high-frequency level. This is standard behaviour for many viscoelastic systems and is characteristic of first-order systems.

Standard time-domain tests such as a creep test or a relaxation test include the application of a step input, which results in an exponential decay in the response from the system. However, there are materials that show a power-law time decay of the form $t^{-\alpha}$ where $\alpha$ varies between 0 and 1, which implies a transfer function with fractional powers of $s$ corresponding to fractional derivative operators [1,2]. For these materials, the phase characteristic does not follow patterns such as those in Figure 2 but instead shows a constant phase, intermediate between zero and $\pi/2$ radians over a wide frequency range.

One of the first recorded observations of power-law decay was when monitoring the behaviour of cheese as it ripens during the maturation process [3]. Since then, there have been many applications where fractional behaviour of materials has been studied [4–6]. In these cases, modelling has involved the incorporation of a "springpot", an element with an impedance described by

$$Z_{springpot} = ks^{\alpha} \tag{3}$$

As the name suggests, this element has a characteristic somewhere between an elastic spring and a viscous dashpot where $k$ is a constant and $\alpha$ describes the fractional order. A zero value for $\alpha$ indicates pure elasticity while a value of one implies purely viscous behaviour. The springpot is typically added to a conventional model such as a standard linear solid (SLS) [7,8], replacing either a spring or a dashpot.

It has been shown mathematically that fractional order systems can be related to a fractal structure [9,10]. For materials, this is relevant because chemical bonds forming elastic connections in

a viscous background can create self-similar structures within the material and, hence, give rise to fractional behaviour.

In this paper, a fractal structure is developed to represent the dynamic behaviour of phase changing materials. The fractal topology is then related to a technique [11] used for modelling fractional systems. Finally, some examples of physical transformations are considered where the aforementioned technique is used to model material phase transformation. Liquid/solid transformations are considered, which raises the issue of gelation, which will be considered in Section 5, when discussing results.

## 2. Fractal and Fractional Modelling

For simple elastic materials, a Hookean elastic model is often used where the stress is solely dependent on a constant, the elasticity, multiplied by the strain. For purely viscous materials, the stress is dependent on the viscosity multiplied by the strain rate. For both these models, the material is considered to be uniform and without structure. These models can be combined to produce simple structural models, termed viscoelastic models such as those in Figure 1. These models are based on long length scales and result in integer order descriptions of the material dynamic response. If one includes more detailed structural topology when using springs and dashpots to model materials at small length scales, non-integer order behaviour can be included. One technique to do this is to use fractal patterns [9,10]. Figure 3 shows a simple fractal structure consisting of springs and dashpots and how it can be condensed to an equivalent impedance $X$.

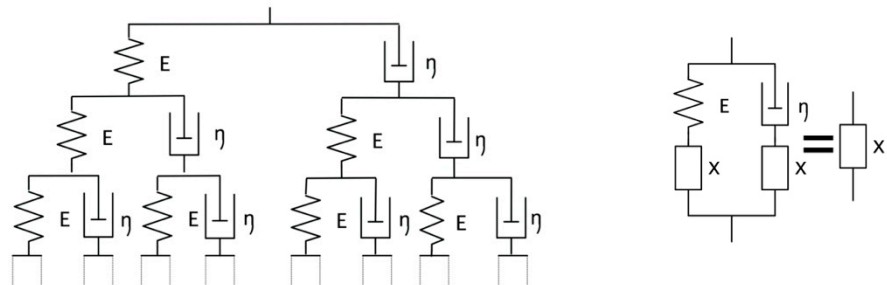

**Figure 3.** Fractal network and equivalent system.

Looking at the structures in Figure 3, one can write

$$X = \frac{1}{\frac{1}{E} + \frac{1}{X}} + \frac{1}{\frac{1}{\eta s} + \frac{1}{X}} \tag{4}$$

so that

$$X = (E\eta s)^{0.5} \tag{5}$$

This gives rise to a fixed fractional order of 0.5. For phase changing materials, the fractional power changes as the transformation progresses, requiring a more sophisticated model. A recursive fractal model has been developed [12] where, depending on the fractal structure, any fractional power value can be achieved. However, it was recognised that visualisation of the physical structure is obscured due to the recursive approach. Here, an iterative/recursive technique is used to establish fractal structures that are somewhat easier to understand and that can be related to the physical phenomenon seen in practice.

To begin this analysis, the impedance derived in Equation (5) will be termed $Z_4(s)$; the use of 4 as a subscript will become apparent later. Now, consider that all the elastic elements in Figure 3 are

replaced by an element of type $Z_4(s)$. Using the same approach to derive Equation (5), we can then find a new element $Z_3(s)$ such that

$$Z_3(s) = \frac{1}{\frac{1}{Z_4(s)} + \frac{1}{Z_3(s)}} + \frac{1}{\frac{1}{\eta s} + \frac{1}{Z_3(s)}} = E^{0.25}(\eta s)^{0.75} \tag{6}$$

Iterating once again by replacing the element $Z_4(s)$ in the network described by Equation (6) by the element $Z_3(s)$, we can define a new fractal pattern $Z_2(s)$ such that

$$Z_2(s) = \frac{1}{\frac{1}{Z_3(s)} + \frac{1}{Z_2(s)}} + \frac{1}{\frac{1}{\eta s} + \frac{1}{Z_2(s)}} = E^{0.125}(\eta s)^{0.875} \tag{7}$$

Iterating once again,

$$Z_1(s) = \frac{1}{\frac{1}{Z_2(s)} + \frac{1}{Z_1(s)}} + \frac{1}{\frac{1}{\eta s} + \frac{1}{Z_1(s)}} = E^{0.0625}(\eta s)^{0.9375} \tag{8}$$

On the other hand, one could take the fractal network in Figure 3 and replace the viscous element with an element of type $Z_4(s)$ to find a new fractal network of type $Z_5(s)$

$$Z_5(s) = \frac{1}{\frac{1}{E} + \frac{1}{Z_5(s)}} + \frac{1}{\frac{1}{Z_4(s)} + \frac{1}{Z_5(s)}} = E^{0.75}(\eta s)^{0.25} \tag{9}$$

Now, iterating this procedure again by replacing the element $Z_4(s)$ in the network described by Equation (9) by the element $Z_5(s)$, we can define a new fractal pattern $Z_6(s)$ such that

$$Z_6(s) = \frac{1}{\frac{1}{E} + \frac{1}{Z_6(s)}} + \frac{1}{\frac{1}{Z_5(s)} + \frac{1}{Z_6(s)}} = E^{0.875}(\eta s)^{0.125} \tag{10}$$

This can be repeated to give

$$Z_7(s) = E^{0.9375}(\eta s)^{0.0625} \tag{11}$$

The process of phase change from liquid to solid could be continued to give ever smaller fractional increments, but if one assumes a start point with a purely viscous system

$$Z_0(s) = \eta s \tag{12}$$

and a finish point with a purely elastic system

$$Z_8(s) = E \tag{13}$$

then, the intervening stages can be described by $Z_1(s)$ to $Z_7(s)$.

If one regards the elastic elements as representing chemical bonds, the material transitions from a weakly connected system dominated by viscous behaviour to one that is highly connected having many chemical bonds and is dominated by an elastic response. From $Z_1(s)$ to $Z_4(s)$, fractal patterns consolidate to form elastic elements, while, from $Z_4(s)$ to $Z_7(s)$, further fractal components account for crosslinking between molecular chains.

The transformation process can be defined by taking the end state $Z_8(s)$ and dividing it by the start state $Z_0(s)$ to define a transition function $H_T$ [11] so that

$$H_T(s) = \frac{E}{\eta s} \tag{14}$$

Then, to determine the state of the system at any stage, the transition function raised to a fractional power $\beta$ is multiplied by the start condition

$$Z(s,\beta) = \left(\frac{E}{\eta s}\right)^{\beta} \times \eta s = (\eta s)^{1-\beta} E^{\beta} \tag{15}$$

Plotting the powers of $s$ as the transition occurs through stages 0 to 8, one obtains the points shown in Figure 4.

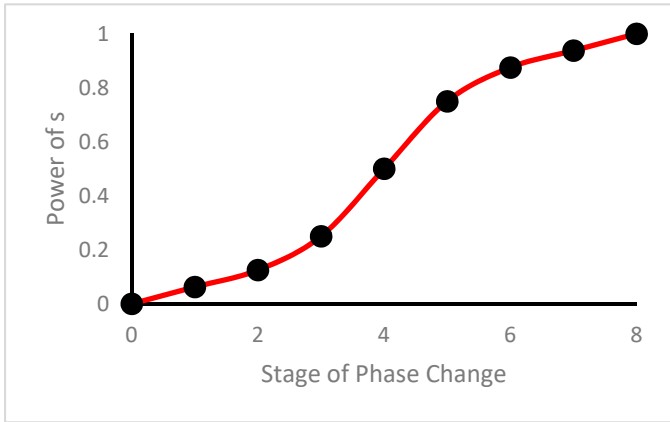

**Figure 4.** Power of $s$ in fractal evolution.

In practice, it would not be expected that the material phase changes all occur in perfect synchrony. To account for this and still allow for a fractal pattern, it is more realistic to consider a random fractal pattern resulting from a distribution of mechanical impedances which shift as the transformation progresses. The continuous gradual change can then be represented by the red trend line in Figure 4.

This trend is often seen in the cure characteristics of adhesives. An example of this can be seen in Figure 5 where the strength of adhesive bonds is presented as the cure develops for a methacrylate adhesive at different bond gaps.

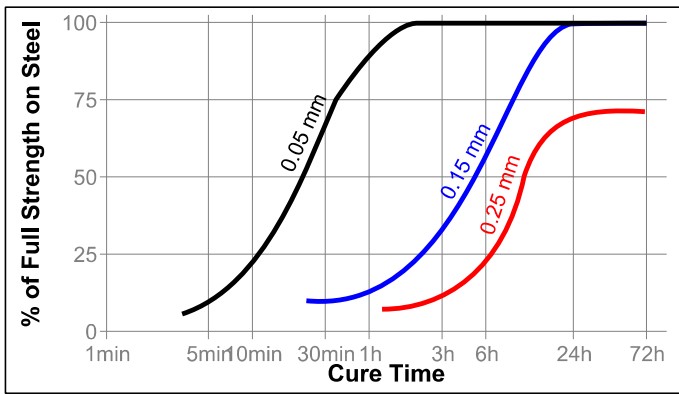

**Figure 5.** Cure speed of adhesive versus bond gap [13].

It seems reasonable to assume the strength would be related to the degree of phase change. Figure 6 shows the fractional power used to model experimental results for a cyanoacrylate adhesive as it cures [11].

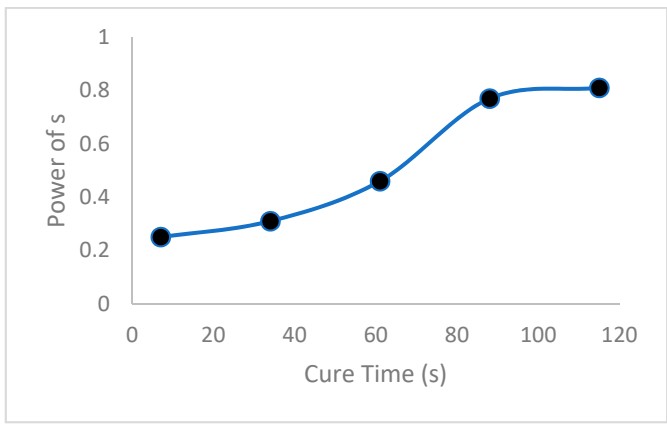

**Figure 6.** Power of *s* in adhesive cure.

Although the start and end values differ from the idealised case in Figure 4, the shape is similar and can be described by a sigmoid function as follows:

$$\beta(t) = \beta_s + \frac{\beta_f - \beta_s}{\left(1 + e^{c(\tau - t)}\right)} \tag{16}$$

where $\beta_s$ is the fractional value at the start of the process, $\beta_f$ is the final value, $\tau$ is the stage of the process at the median value of $\beta$ and $c$ is a constant. The independent variable can be time or some other parameter such as temperature depending on the physical process involved.

The analysis so far results in a springpot of order $\beta$, which, as mentioned in the Introduction, is the element used in conjunction with springs and dashpots to model systems showing fractional behaviour. The fractal model presented in Section 2 assumes a perfectly viscous material transitioning to a perfectly elastic model during the phase change process, resulting in a dashpot with varying fractional power. In practice, materials will often show various viscoelastic properties before and after phase change. To account for such behaviour, a more general approach may be used [11], whereby the start and end state used to formulate the transition function in Equation (14) can be any appropriate integer order viscoelastic function such as a Maxwell or Kelvin Voigt element rather than just a purely viscous or purely elastic one. Thus, by inspecting the bode response of the material before and after transformation, the viscous and elastic elements can be substituted by the appropriate viscoelastic models to generate a transition function appropriate to the behaviour shown in the experimental data. This allows the modeller to create the overall description of the transformation process $Z(s, \beta)$ based on some understanding of the physical characteristics of the material.

## 3. Test Methods and Materials

The dynamic mechanical analysis of liquids and soft solids is carried out using the Micro Fourier Rheometer (GBC Scientific). It is an oscillatory squeeze film rheometer that uses an axisymmetric geometry where flat circular plates of radius R move relative to one another along their axes (z-axis) with amplitudes up to 20 μm. A schematic of the test geometry is shown in Figure 7.

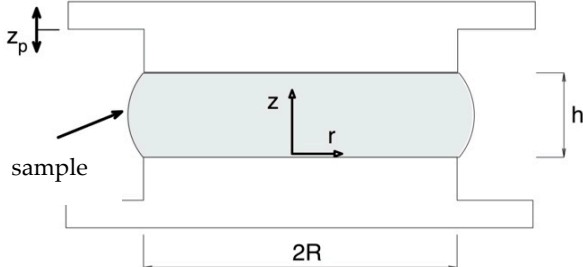

**Figure 7.** Schematic of squeeze film geometry showing a sample under test (not to scale).

The top plate moves with displacement $z_p$, squeezing/stretching the material between it and the fixed lower plate. The displacement in the upper plate is measured to calculate the instantaneous height h, and the force induced in the lower plate is recorded from which the complex modulus of the sample under test is determined.

One of the issues when testing phase changing materials is that they will eventually bond the rheometer together, rendering it unusable. To preserve the integrity of the rheometer, a removable system using neodymium magnets is employed to hold the top and bottom plates in place. A representation of the top plate is shown in Figure 8 and a similar setup is used for the bottom plate. The forces incurred during testing are well below the magnetic holding force on the removable plate, but once testing is finished, it is then possible to slide the bonded removable plates out from the rheometer and separate them.

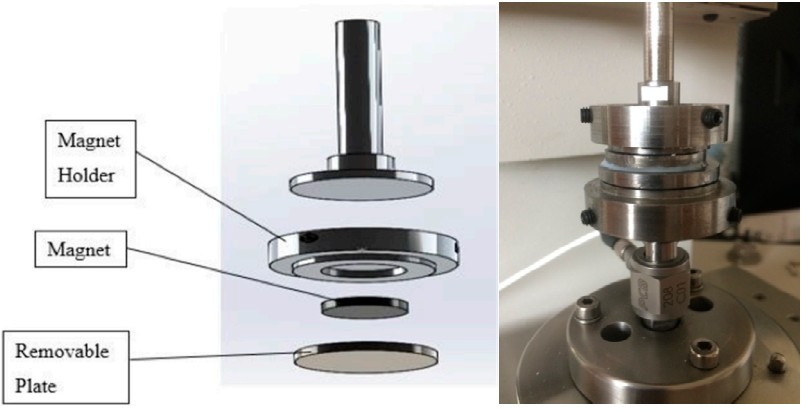

**Figure 8.** Design of top plate (**left**) and entire test assembly (**right**).

The force and displacement data are Fourier-transformed and analysed using a non-linear spectral technique [14] so that the material characteristics can be extracted in terms of complex modulus and phase. It is important to acquire the data quickly so that the transient nature of the material properties can be treated as being quasi-static. For this reason, a band-limited random signal between 0 and 20 Hz is used, which allowing for a 40 Hz sample rate, and 400 points requires 10 s to acquire. This is comparable to the high-speed techniques described in [15] when using the swept sine technique. The spectral data are calculated using just one record rather than the usual ensemble average technique due to the transient nature of the material. Data at low frequencies are somewhat compromised by the piezoelectric force transducer, which is evident in the results presented.

## 4. Results

Three examples of phase changing materials are examined, each with a different mechanism causing the change. The three materials and the mechanisms are:

(i)    Dairy cream thickening over time due to shearing, termed rheopexy.

(ii)  Water/gelatine solidifying due to chemical bonding.

(iii)  Ethyl vinyl acetate (EVA) melting with temperature.

Various models are proposed for each of these processes using the strategy described in Section 2, and parameter values are determined using the generalised reduced gradient solver in Excel.

### 4.1. Cream

Fresh dairy cream was placed between plates with a 100 μm gap and subjected to random excitation with an amplitude of up to 10 μm over a period of 30 min. From the observed spectra, a transition function based on a viscous ($\eta s$) to Maxwell $\left(\frac{E_1\eta_1 s}{E_1+\eta_1 s}\right)$ transfer function was assumed. This produces a transition function

$$H_T(s) = \left(\frac{E_1\eta_1 s}{E_1 + \eta_1 s}\right)/(\eta s) \tag{17}$$

so that the material impedance becomes

$$Z(s,\beta) = (\eta s)^{1-\beta}\left(\frac{E_1\eta_1 s}{E_1 + \eta_1 s}\right)^\beta \tag{18}$$

The measured frequency response function and modelled version is presented in terms of magnitude and phase for whipping times of 0, 20 and 30 min in Figure 9.

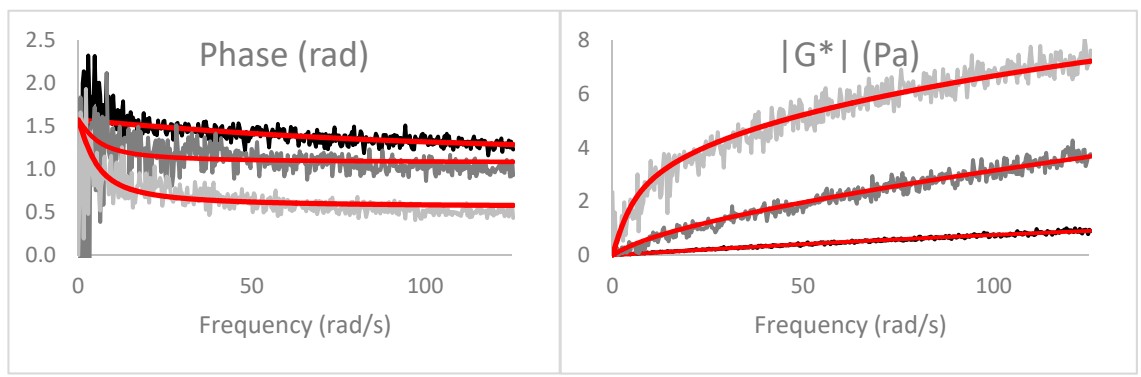

**Figure 9.** Frequency response for Cream at 10 ▬, 20 ▬ and 30 ▬ min (experimental), and the modelled impedance ▬▬.

The parameter values for the modelled data are shown in Table 1.

**Table 1.** Model values for viscous to Maxwell transformation of cream.

| Time (min) | 0 | 20 | 30 |
|---|---|---|---|
| $\eta$ (Pas) | 0.006 | 0.042 | 2.276 |
| $E_1$ (Pa) | 1.90 | 1.70 | 1.00 |
| $\eta_1$ (Pas) | 0.023 | 0.297 | 0.193 |
| $\beta$ | 0.293 | 0.320 | 0.651 |

### 4.2. Gelatine

A water/gelatine mix (100 mL water to 3 g of Knox Gelatine) was prepared by mixing room-temperature water with gelatine until the gelatine was dissolved. The solution was then heated to 40 °C over a period of 1 min. A small amount of the mixture was then placed between plates with a 200 μm gap and subjected to a 1 μm random excitation over a period of 55 min. From the

observed spectra, it is convenient to select a transition function based on a Kelvin Voigt $(E + \eta s)$ to elastic $E_1$ transfer function.

$$H_T(s) = E_1/(E + \eta s) \tag{19}$$

Thus, the material impedance becomes

$$Z(s, \beta) = (E + \eta s)^{1-\beta} E_1{}^{\beta} \tag{20}$$

The measured frequency response function and modelled version are presented in terms of magnitude and phase at 0, 4 and 9 min in Figure 10. Finer time resolution is used to identify the gel point as displayed in Figure 11 which shows values at 0, 1 and 2 min.

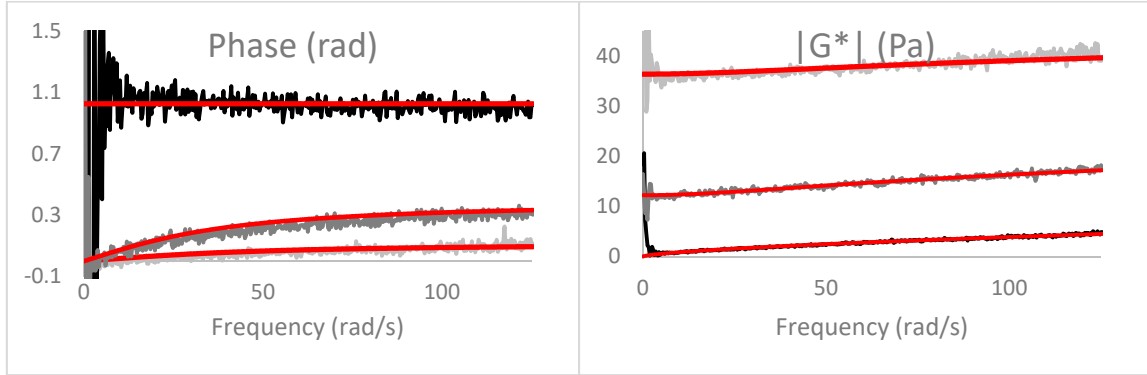

**Figure 10.** Frequency response for gelatine at 0 ▬, 4 ▬ and 9 ▬ min (experimental), and the modelled impedance ▬.

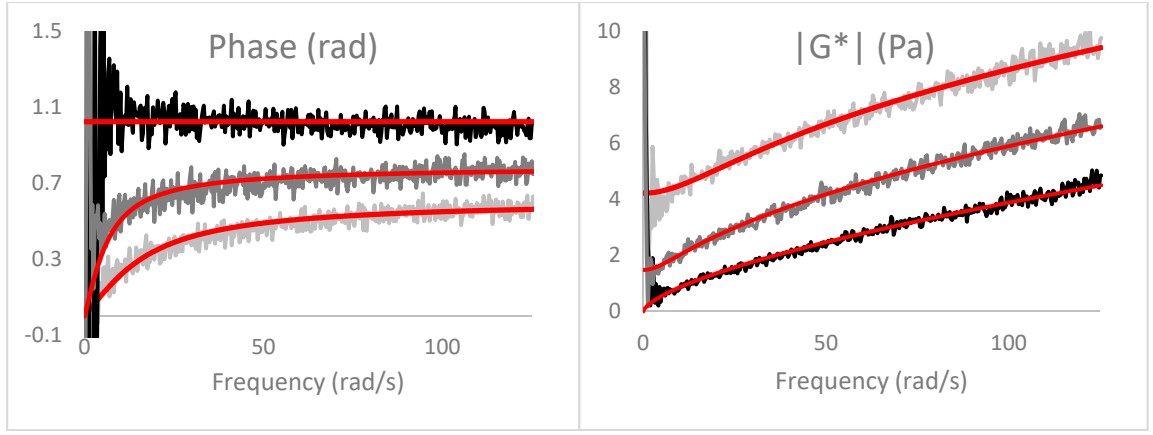

**Figure 11.** Frequency response near the gel point for gelatine at 0 ▬, 1 ▬ and 2 ▬ min (experimental), and the modelled impedance ▬.

The fitted data in Table 2 indicate that the first data point is, in fact, a viscous to elastic model as the value of $E$ is zero for this point.

**Table 2.** Model values for Kelvin Voigt to elastic transformation of gelatine.

| Time (min) | 0 | 1 | 2 | 4 | 9 | 30 | 55 |
|---|---|---|---|---|---|---|---|
| $E$ (Pa) | 0.00 | 0.96 | 2.88 | 14.12 | 23.54 | 12.79 | 9.23 |
| $\eta$ (Pas) | 0.075 | 0.151 | 0.178 | 0.418 | 0.559 | 0.808 | 1.032 |
| $E_1$ (Pa) | 1.1 | 2.3 | 5.4 | 11.6 | 37.7 | 100.1 | 134.7 |
| $\beta$ | 0.348 | 0.499 | 0.610 | 0.746 | 0.925 | 0.986 | 0.990 |

### 4.3. EVA

EVA adhesive was placed between heated plates and heated from 36 °C to 97 °C using a 1000 μm gap and subjected to a 1 μm random excitation. From the observed spectra, two transition functions were deemed necessary to model the material impedance. The first describes an elastic $E$ to Kelvin Voigt $(E_1 + \eta_1 s)$ transition and the second describes a Maxwell $\left(\frac{E\eta s}{E+\eta s}\right)$ to viscous $(\eta_1 s)$ transition.

Phase 1 elastic to Kelvin Voigt:

$$H_T(s) = (E_1 + \eta_1 s)/E \tag{21}$$

thus, the material impedance becomes

$$Z(s, \beta) = E^{1-\beta}(E_1 + \eta_1 s)^{\beta} \tag{22}$$

Phase 2 Maxwell to viscous:

$$H_T(s) = (\eta_1 s)\bigg/\left(\frac{E\eta s}{E + \eta s}\right) \tag{23}$$

thus, the material impedance becomes

$$Z_1(s, \beta) = \left(\frac{E\eta s}{E + \eta s}\right)^{1-\beta} (\eta_1 s)^{\beta} \tag{24}$$

The measured frequency response function and modelled version are presented in terms of magnitude and phase at 36 °C, 72 °C and 97 °C in Figure 12.

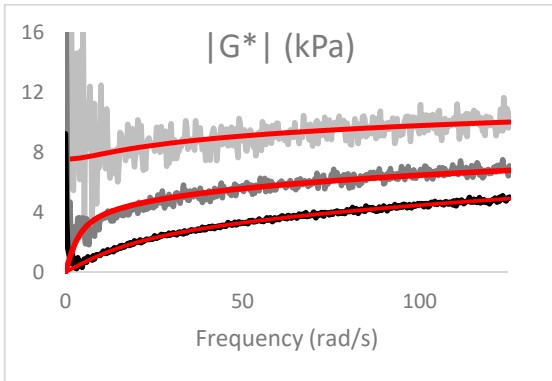 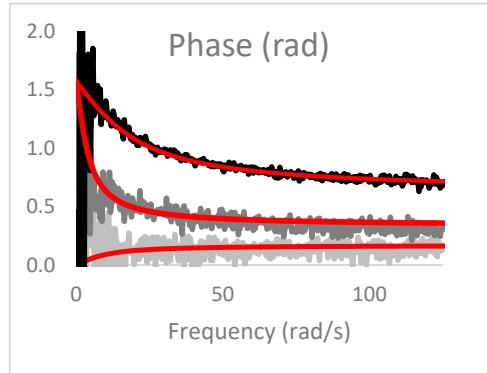

**Figure 12.** Frequency response for ethyl vinyl acetate (EVA) at 97 °C ▬, 72 °C ▬ and 36 °C ▬ (experimental), and the modelled impedance ▬.

Due to ambiguity of the results at 72 °C, experimental results at 64 °C, 72 °C and 79 °C are presented in Figure 13 (see Section 5).

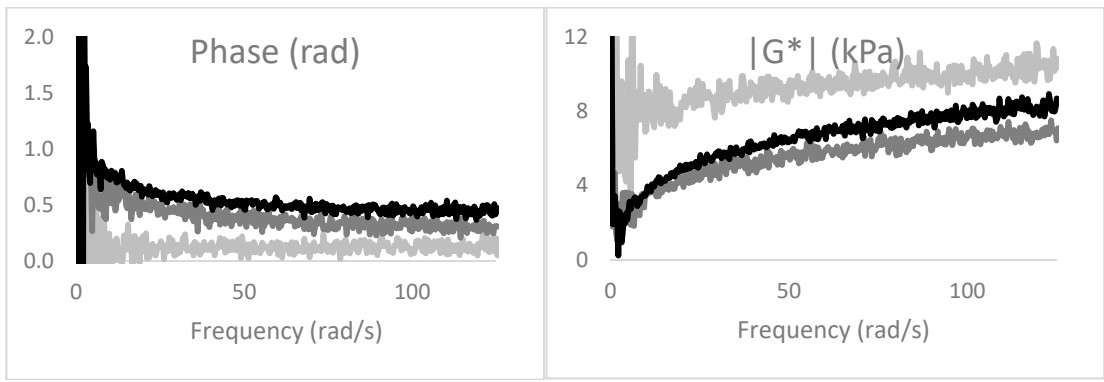

**Figure 13.** Experimental frequency response for EVA near the melting point, at 79 °C ━, 72 °C ━, 64 °C ━.

Parameter values for phase 1 (solid) and phase 2 (liquid) are presented in Tables 3 and 4, respectively.

**Table 3.** EVA elastic to Kelvin Voigt transition.

| EL–KV | 36 | 52 | 64 | 72 |
|---|---|---|---|---|
| $E$ (kPa) | 16.7 | 16.0 | 16.4 | 16.9 |
| $E_1$ (kPa) | 0.011 | 0.013 | 0.016 | 0.005 |
| $\eta_1$ (Pas) | 0.0012 | 0.0011 | 0.0016 | 0.0039 |
| $\beta$ | 0.108 | 0.112 | 0.106 | 0.250 |

**Table 4.** EVA Maxwell to viscous transition.

| Temp °C | 72 | 79 | 84 | 88 | 91 | 93 | 95 | 97 |
|---|---|---|---|---|---|---|---|---|
| $E$ (kPa) | 0.33 | 0.35 | 0.29 | 0.31 | 0.29 | 0.30 | 0.30 | 0.31 |
| $\eta$ (Pas) | 0.08 | 0.070 | 0.043 | 0.025 | 0.023 | 0.020 | 0.017 | 0.016 |
| $\eta_1$ (Pas) | 5.8 | 4.0 | 2.4 | 1.8 | 1.8 | 1.6 | 1.5 | 1.5 |
| $\beta$ | 0.212 | 0.305 | 0.364 | 0.390 | 0.388 | 0.397 | 0.399 | 0.399 |

## 5. Discussion of Results

### 5.1. Cream

From the bode plots for cream (Figure 9), a viscous model was chosen for the start point because there was no yield point evident in the magnitude plot ($|G^*| = 0$ at 0 rad/s), and the phase tended to $\pi/2$ radians at 0 rad/s at all stages during the transformation process. A Maxwell model was selected instead of a purely elastic model for the end point because the phase was not constant across all frequencies, indicating the need for the more sophisticated model. Beyond 30 min, it was noted that the modulus began to reduce again and so testing was stopped.

The modelled results match experimental results well for both the magnitude and phase plots, albeit that the noise in the experimental data is not captured by the model, which is to be expected. More noise was present as the magnitude increased for all three processes examined because the displacement amplitude was decreased during testing in these instances to minimise stress-induced disturbances. The reduced amplitude, therefore, had a higher signal-to-noise ratio, which is apparent in the experimental results.

As the dynamic strain progresses, the complex modulus increases. The phase reduces as the material becomes more elastic, and this corresponds to $\beta$ increasing as the transition from the viscous to Maxwell behaviour develops. The increase in $\beta$ is quite marked between 20 and 30 min, indicating a rapid change in the rheology, a phenomenon that is typical when whipping cream. The other parameters in the model also change, indicating that the start and end points do not stay constant,

indicating an affine transformation (Table 1). Studies have shown that the whipping process of cream occurs due to the inclusion of air bubbles, which give rise to elasticity [16]. The inclusion of air means that the fluid must move around these bubbles and is subjected to higher shear rates, thereby increasing the apparent viscosity.

*5.2. Gelatine*

The frequency response functions for this material are shown in Figure 10 for early (0 min), middle (4 min) and late (9 min) stages of the solidification process. Although a Kelvin Voigt to elastic model transition was chosen to model gelatine, the first data point is represented by a viscous to elastic model as the parameter $E$ is zero for this point (Table 2). The model at zero minutes is effectively that of Equation (15) and has liquid-like behaviour. Beyond this point, at 1 min, the material does possess a yield strength (non-zero magnitude at 0 rad/s) and the Kelvin Voigt to elastic model applies. For the modelling approach here to capture yield behaviour, neither the start nor the end condition can have a liquid nature, otherwise, an $s$ term appears in the numerator of the material impedance, corresponding with zero magnitude for the complex modulus $|G^*|$ at 0 rad/s. To focus on the point of gelation more closely, the frequency response functions at 0, 1 and 2 min are shown in Figure 11. Here, it is apparent that the onset of solidification is around 1 min, while, before this point, gelatine behaves more like a liquid.

At the second data point, at 1 min, a yield stress is present (i.e., $E \neq 0$), at which point $\beta$ is 0.499. By the end of the process at 55 min, $\beta$ is very close to unity (0.99) and the material is almost completely elastic in nature.

Plotting the evolution of $\beta$ in Figure 14, the shape follows the later stages of the sigmoid curve seen in Figure 4. Given that the solution is mainly water, one could expect the behaviour to be almost entirely viscous at the start with $\beta$ close to zero. However, due to the method of preparation, some time elapses before the mixture of gelatine can be tested, by which stage the gelling process is already underway. In this case, the point of gelation is considered to occur around 1 min, at which point $\beta$ is 0.499. Beyond this, the $\beta$ value rises very close to 1. Thus, at gelation, the $\beta$ value is very close to the median. The sigmoid equation for the evolution of $\beta$ for gelatine is given in Equation (25).

$$\beta(T) = \frac{1}{\left(1 + e^{0.5(1-t)}\right)} \tag{25}$$

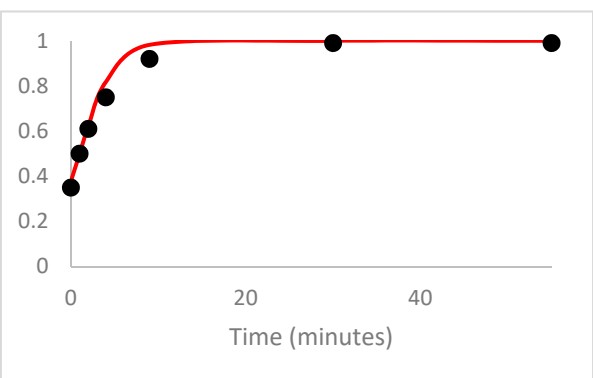

**Figure 14.** Evolution of $\beta(t)$ for gelatine.

There is much debate about solidification and how it is defined, which, in rheological terms, amounts to specifying the critical point of gelation [17]. Broadly speaking, a solid can maintain a yield stress while a liquid cannot. In rheological studies, a simple approach has been to look for the cross-over point where the storage modulus $G'$ (real component of the complex modulus), which defines elastic behaviour, begins to exceed the loss modulus $G''$ (complex component of the complex modulus),

which defines viscous behaviour. A more sophisticated approach relies on the observation that, at and beyond the gel point, the real and complex moduli run parallel to each other at low frequencies, so that $G'/G''$ is constant corresponding to a constant phase. This implies fractional derivative behaviour, which is linked to fractal structures, as described in Section 1. However, in this paper, it is proposed that a fractal structure is present from the start, that is, prior to the critical gel point where the ratio of $G'/G''$ is not constant. In addition, if the presence of a yield point is used to define the point of gelation, then a discrete change in material structure during modelling is required with the exclusion of liquid-like models such as purely viscous or Maxwell models at and beyond the gel point.

*5.3. EVA*

The magnitude plot (Figure 12) for EVA has a non-zero value at low temperatures, indicating solid-like behaviour, while, at higher temperatures, no yield stress is apparent and the material behaves as a liquid. As a consequence, it is necessary to use two regimes when modelling the behaviour during the heating process. In the first, from 36 °C to 64 °C, an elastic to Kelvin Voigt transition is assumed, and from 72 °C to 97 °C, a Maxwell to viscous transition is used (Tables 3 and 4, respectively). The experimental results for 72 °C are somewhat ambiguous and the evolution of material behaviour at 64 °C, 72 °C and 79 °C is shown in Figure 13. It should be remembered that the data at the very lowest frequencies are affected by the low frequency limit of the piezoelectric force transducer and cannot be relied upon below 3 rad/s. Interpreting material behaviour, therefore, relies on studying the trends towards low frequencies. Looking at the phase first, there is an increase in phase as the material heats up and transforms from solid-like to liquid-like behaviour. The trends in the magnitude plots towards low frequencies indicate that there is a yield point at 64 °C and 72 °C but not 79 °C. Yet, at 79 °C the magnitude plot increases above that for 72 °C as frequency increases. This shows that the viscous behaviour at 79 °C becomes dominant in the liquid-like state at this temperature.

At 72 °C, the plots of magnitude and phase show non-zero magnitude at 0 rad/s and, yet, do not exhibit zero phase at this frequency, therefore displaying both solid-like and liquid-like features, and are difficult to classify. Figure 15 shows elastic to Kelvin Voigt (solid-like) and Maxwell to viscous (liquid-like) models fitted to the data at this temperature.

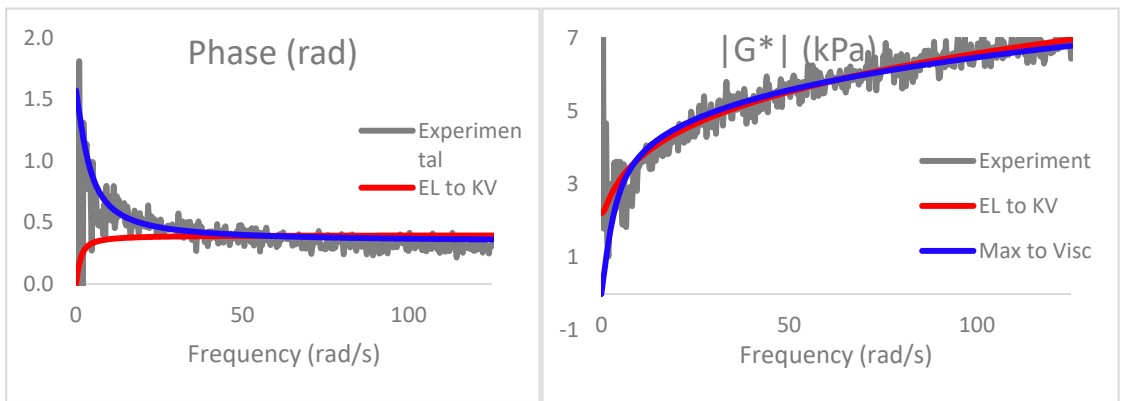

**Figure 15.** Model fits to experimental data at 72 °C.

The elastic to Kelvin Voigt model can maintain a yield stress but does not represent the phase well. By contrast, the Maxwell to viscous model is more accurate in terms of phase but has no yield behaviour. For Figure 12, the Maxwell to viscous model was used; however, the behaviour is complex and possibly points towards plastic behaviour where, near the gel point, elastic bonds are being broken and reformed during the testing process.

From Figure 16, it is apparent that for much of the solid phase, $\beta$ is approximately 0.1 after which there is a rapid increase at 72 °C and 79 °C before $\beta$ settles to around 0.4. The sigmoid equation describing the evolution of $\beta$ as a function of temperature is given in Equation (26).

$$\beta(T) = 0.1 + \frac{0.3}{\left(1 + e^{0.22(75-T)}\right)} \tag{26}$$

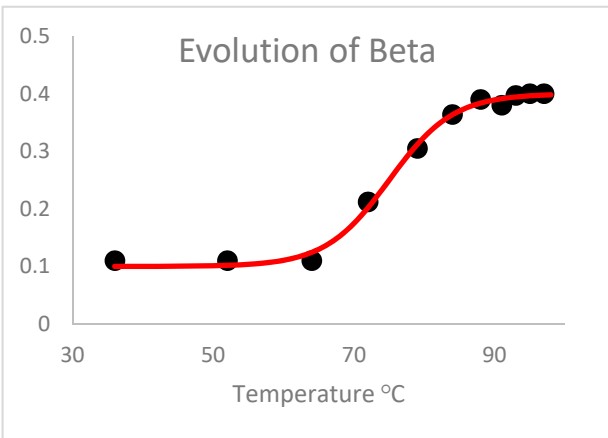

**Figure 16.** Evolution of $\beta(T)$ for EVA.

The midpoint value for $\beta$ is 0.25 at 75 °C, and it is beyond this point at 79 °C where the material loses any yield behaviour, and it could be argued that the gel point corresponds to the median value for $\beta$ between 0.1 and 0.4.

## 6. Conclusions

A fractal topology has been used to describe the rheological phase transition in materials. This results in a sigmoidal-shaped evolution of material properties that is often seen in practise. The fractal structure can be related to a fractional derivative modelling approach that has been used previously when characterising adhesive cure and which is used here to model the whipping process of cream, the solidification of gelatine and the melting of EVA. The three materials were chosen to represent three different mechanisms of phase transition, rheopexy for whipping cream, chemical bonding for the solidification of gelatine and temperature-induced melting for EVA. Using the fractional derivative modelling approach, a good fit was obtained when modelling most of the material's dynamic response during phase transition.

For solid-like behaviour, the modelling approach adopted here requires start and end points that behave as solids. Once a liquid-type model is incorporated at either or both of the terminal points, the overall system will not be able to sustain a yield condition and will, therefore, have liquid-like behaviour. When dealing with liquid/solid transitions, a non-zero magnitude of the frequency response function at 0 rad/s was used to indicate the onset of gelation. In the case of the melting process for EVA, it was not possible to accurately capture both the magnitude and the phase with the fractional model. This arose because the material exhibits a yield stress in terms of magnitude and, yet, does not have zero phase at 0 rad/s.

**Funding:** This research received no external funding.

**Acknowledgments:** This paper is based upon work from COST Action Fractional (CA15225), supported by COST (European Cooperation in Science and Technology).

**Conflicts of Interest:** The authors declare no conflict of interest.

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
