# Peer review of "Fractal and Fractional Derivative Modelling of Material Phase Change"

_fractalfract, doi:10.3390/fractalfract4030046_

Round 1
Reviewer 2 Report
This paper exploits the connection between non-local time behaviors in generalized viscoelastic models. From this viewpoint a recursive/iterative methodology is presented to model phase change —a transition from a viscous to an elastic state.
This work builds on the recent work from the author (ref 7) that develops and uses a similar technique for characterizing adhesive cure. In the current work the proposed methodology is experimentally tested against three other materials.
This paper is very well written, making it easy to grasp insightful ideas about material phase change. It represents an advance over the previous work, ref 7, be showing that the model can be extended to other viscoelastic materials.
I recommend publication as is.
Just two comments for the author to consider.
On line 52. Why is t^{-\alp} referred to a as an algebraic as opposed to power-law decay?
How might the ideas in this paper relate to other phase change models, e.g., phase filed models of solid to liquid phase change?—see
Phase-field simulation of solidification
WJ Boettinger, JA Warren, C Beckermann, A Karma Annual review of materials research 32 (1), 163-194
Author Response
Dear Reviewer 2,
Thank you for your review of the paper.
The term power law has been substituted for algebraic.
After reviewing the suggested Phase Field paper, I think that the similarity lies in the fact that a single parameter is used to represent material phase state.
In the current paper, a link is made between material phase and the fractional power which is strongly related to the spectral phase characteristic. The fractional power is shown to be related to the fractal topology and therefore related to spatial frequency. Underpinning the analysis is the stress strain relationship which one could link to kinetics and thermodynamic considerations.
In the phase field paper, the material phase is determined using the value phi. Phi is a function of position and time and is derived from thermodynamic principles. The formulation allows one to model the position of the material phase (liquid/solid) boundary.
The current paper does not include position as a variable and assumes that the behaviour is homogeneous in a spatial sense. Thus while the papers are related there does not seem to be a mathematical correlation between them.
Regards
Harry
Reviewer 1 Report
This paper applies transfer function methods to derive and experimentally validate fractional order viscoelasticity. The results compares theory to experiments which is rarely done and should be commended. I just have a few minor points to consider.
There's quite a bit of important results in the field of fractals and fractional models I recommend the authors consider referencing. Key references include:
B. J. West and P. Grigolini, Complex webs: Anticipating the improbable. Cambridge University Press, 2010.
V. E. Tarasov, “Review of some promising fractional physical models,” International Journal of Modern Physics B, vol. 27, no. 09, p. 1330005, 2013.
S. Mashayekhi, M. Y. Hussaini, and W. Oates, “A physical interpretation of fractional viscoelasticity based on the fractal structure of media: Theory and experimental validation,” Journal of the Mechanics and Physics of Solids, vol. 128, pp. 137–150, 2019.
F. Mainardi, “An historical perspective on fractional calculus in linear viscoelasticity,” Fractional Calculus and Applied Analysis, vol. 15, no. 4, pp. 712–717, 2012.
R. L. Bagley and P. Torvik, “A theoretical basis for the application of fractional calculus to viscoelasticity,” Journal of Rheology (1978-present), vol. 27, no. 3, pp. 201–210, 1983.
The authors introduce dynamics of phase between an input and output in Figure 2. I recommend clarifying this phase up-front so there is no confusion with "material phases" which is discussed later.
The authors use a series of first order dashpots in a fractal series to construct and effective fractional order system. How can you make an assumption of a local integer order viscous dashpot to create a fractional order model? How may this be physically motivated and how are the length scales of such structures homogenized into a phenomenlogical framework?
The authors chose between Kelvin Voigt and Maxwell models based on a priori knowledge of the spectra. How can this be justified?
Figure 11 is introduced on page 8, but not discussed until much later.
Middle of page 9, Figure 11 is referenced, but I believe it should have been Figure 12.
Author Response
Dear Referee 1,
Thank you for your review of the paper. In relation to your comments, I make the following responses:
- I have included references 2-5 in your list. I did not include the first reference because it was not specific to the subject matter covered in the paper.
- I have included the following text in the paper -
It is worth mentioning here that the phase in a bode plot is the frequency equivalent of the time shift between the output signal when compared to the input signal as the signal passes through a system. This is not to be confused with the term phase used to describe the physical form of a material, for example liquid or solid.
- I have included the following text in the paper -
For simple elastic materials, a Hookean elastic model is often used where the stress is solely dependent on a constant, the elasticity, multiplied by the strain. For purely viscous materials the stress is dependent on the viscosity multiplied by the strain rate. For both of these models, the material is considered to be uniform and without structure. These models can be combined to produce simple structural models, termed viscoelastic models such as those in Figure 1. These models are based on long length scales and result in integer order descriptions of the material dynamic response. If one includes more detailed structural topology when using springs and dashpots to model materials at small length scales, non-integer order behaviour can be included. One technique to do this is to use fractal patterns [9,10]. Figure 3 shows a simple fractal structure consisting of springs and dashpots and how it can be condensed to an equivalent impedance X .
- The following text has been added -
The analysis so far results in a springpot of order which as mentioned in the Introduction is the element used in conjunction with springs and dashpots to model systems showing fractional behaviour. The fractal model presented in Section 2 assumes a perfectly viscous material transitioning to a perfectly elastic model during the phase change process resulting in a dashpot with varying fractional power. In practice materials will often show various viscoelastic properties before and after phase change. To account for such behaviour, a more general approach may be used [11], whereby the start and end state used to formulate the transition function in Equation 14 can be any appropriate integer order viscoelastic function such as a Maxwell or Kelvin Voigt element rather than just a purely viscous or purely elastic one. Thus by inspecting the bode response of the material before and after transformation, the viscous and elastic elements can be substituted by the appropriate viscoelastic models to generate a transition function appropriate to the behaviour shown in the experimental data. This allows the modeller to create the overall description of the transformation process based on some understanding of the physical characteristics of the material.
-
The results presented in bode charts are discussed in the discussion of results section under headings appropriate to each material tested. This is the case for Figure 11.
-
Reference to Figure 11 was incorrect and has been changed to refer to Figure 12.
Regards
Harry